# Tailoring the coercive field in ferroelectric metal-free perovskites by hydrogen bonding

Hwa Seob Choi[1,7], Shunning Li [2,7], In-Hyeok Park [3], Weng Heng Liew[4], Ziyu Zhu[5], Ki Chang Kwon [1,5], Lin Wang[5], In-Hwan Oh [6], Shisheng Zheng[2], Chenliang Su [1], Qing-Hua Xu [5], Kui Yao [4], Feng Pan [2✉] & Kian Ping Loh [1,5✉]

The miniaturization of ferroelectric devices in non-volatile memories requires the device to maintain stable switching behavior as the thickness scales down to nanometer scale, which requires the coercive field to be sufficiently large. Recently discovered metal-free perovskites exhibit advantages such as structural tunability and solution-processability, but they are disadvantaged by a lower coercive field compared to inorganic perovskites. Herein, we demonstrate that the coercive field (110 kV/cm) in metal-free ferroelectric perovskite MDABCO-NH$_4$-(PF$_6$)$_3$ (MDABCO = N-methyl-N'-diazabicyclo[2.2.2]octonium) is one order larger than MDABCO-NH$_4$-I$_3$ (12 kV/cm) owing to the stronger intermolecular hydrogen bonding in the former. Using isotope experiments, the ferroelectric-to-paraelectric phase transition temperature and coercive field are verified to be strongly influenced by hydrogen bonds. Our work highlights that the coercive field of organic ferroelectrics can be tailored by tuning the strength of hydrogen bonding.

[1] SZU-NUS Collaborative Innovation Center for Optoelectronic Science & Technology, International Collaborative Laboratory of 2D Materials for Optoelectronics Science and Technology of Ministry of Education, Institute of Microscale Optoelectronics, Shenzhen University, 518060 Shenzhen, P.R. China. [2] School of Advanced Materials, Peking University Shenzhen Graduate School, 518055 Shenzhen, P.R. China. [3] Graduate School of Analytical Science and Technology (GRAST), Chungnam National University, Daejeon 34134, Republic of Korea. [4] Institute of Materials Research and Engineering, A*STAR (Agency for Science, Technology and Research), 2 Fusionopolis Way, 138634 Singapore, Singapore. [5] Department of Chemistry, National University of Singapore, 3 Science Drive 3, 117543 Singapore, Singapore. [6] Neutron Science Division, Korea Atomic Energy Research Institute, Daejeon 34057, Republic of Korea. [7] These authors contributed equally: Hwa Seob Choi, Shunning Li. ✉email: panfeng@pkusz.edu.cn; chmlohkp@nus.edu.sg

Since the discovery of ferroelectricity in 1920 in Rochelle salt, enormous research interests have been dedicated to the study of ferroelectrics[1]. Prior to the renaissance of organic ferroelectrics in the past decade, research interests have always been centred on inorganic materials. In contrast to typical inorganic ferroelectric perovskites such as $BaTiO_3$ (BTO) and $Pb(Zr,Ti)O_3$ (PZT)[2,3], the organic counterparts embrace a unique set of appealing attributes, including high flexibility, fracture-resistance, and solution-processability[4,5]. However, the construction of all-organic perovskites is constrained by the types of organic cations and anions that comply with Goldschmidt tolerance factor, which restricts discoveries to a few examples. Among these, the ferroelectric ones are even rarer[6,7]. For example, the recently discovered isostructural family of piperazinium[2+] all-organic perovskites are mostly non-ferroelectric[8,9]. Achieving ferroelectricity in all-organic perovskites is challenging because polar molecular building blocks generally prefer to adopt an antiparallel arrangement in order to minimize the electrostatic energy of the system[10-13]. One recent breakthrough is the discovery of ferroelectric MDABCO-NH$_4$-I$_3$ (MDABCO = N-methyl-N′-diazabicyclo[2.2.2]octonium) perovskite[14], whose spontaneous polarization was reported to be comparable to that of BTO. This organic ferroelectric surpasses many inorganic ferroelectrics in terms of parameters such as electrocaloric strength, Curie-Weiss constant and critical field for phase transition[15]. However, a major drawback of MDABCO-NH$_4$-I$_3$ is its low coercive field (<12 kV/cm). Given that a coercive field larger than 100 kV/cm is required for voltage switching in the range of 1–2 V for a 100 nm thick film[16], the coercive field of MDABCO-NH$_4$-I$_3$ would require a thickness of at least 830 nm to enable operation at 1 V switching-voltage. This presents a roadblock for the miniaturization of the ferroelectric device. On the other hand, a coercive field higher than 1000 kV/cm (in PVDF-TrFE or HfO$_2$) reduces the thickness of the ferroelectric channel to 10 nm, but the drawback is the device is susceptible to a high leakage current, or even dielectric breakdown[17,18]. It is therefore highly desirable to achieve a coercive field on the scale of 100 kV/cm, which demands new design strategies for all-organic ferroelectrics.

Herein, we demonstrate the modulation of the coercive field in organic ferroelectric perovskites via the reinforcement of hydrogen bonding. An all-organic perovskite crystal MDABCO-NH$_4$-(PF$_6$)$_3$ was successfully synthesized, where strong N–H⋯F hydrogen bonds are formed by the N–H groups of MDABCO with the highly electronegative F (Fig. 1) in PF$_6$. The coercive field of this all-organic compound reaches 110 kV/cm and its remnant polarization of 5.7 μC/cm$^2$ agrees with our Berry phase calculated value of 6.5 μC/cm$^2$. The coercive field can be enhanced further to 138 kV/cm by deuteration, which corroborates the fact that hydrogen bonding influences the coercive field. Our study highlights that coercive field in metal-free perovskites can be enhanced by regulating the strengths of intermolecular hydrogen bonds, suggesting that organic ferroelectrics have wide chemical tunability.

## Results and discussion

As a typical organic perovskite, MDABCO-NH$_4$-I$_3$ is composed of divalent amine (MDABCO) at the cuboctahedral centre (A-site), NH$_4^+$ cation at the octahedral centre (B site) and I$^-$ anions occupying the vertices of the octahedron (X site). As MDABCO exhibits the highest remnant polarization among known diamines, we chose it for the A-site cation[14]. To strengthen the hydrogen bond, we replaced I$^-$ with PF$_6^-$ anions (Fig. 2a), the latter is compatible with NH$_4^+$ and MDABCO$^{2+}$, yielding a perovskite structure with a tolerance factor of 0.914 (Table S1).

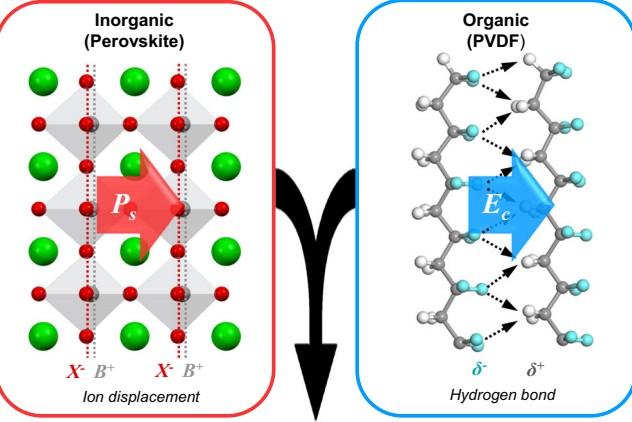

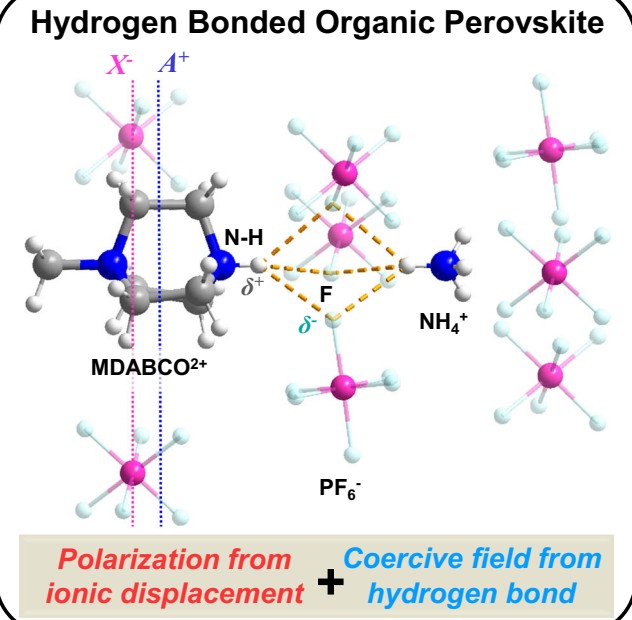

**Fig. 1 Schematic of hydrogen-bond-assisted modulation of polarization and coercive field for all-organic ferroelectric perovskites.** Red box shows ferroelectric polarization in inorganic perovskite due to ion displacement; Blue box shows ferroelectric polarization due to hydrogen bonds in organic ferroelectric PVDF; Black box shows a combination of both ion displacement and hydrogen bonds contribute to ferroelectric polarization in organic perovskite.

MDABCO-NH$_4$-(PF$_6$)$_3$ (denoted as MNP$_3$ hereafter) crystal was grown by the slow evaporation of a mixture with a stoichiometric ratio of MDABCO-(PF$_6$)$_2$ and NH$_4$PF$_6$, where a single-crystal of 5 × 4 × 1 mm size could be obtained (Fig. 2b) Detailed synthesis procedure is described in Supplementary Information.

The structure of MNP$_3$ was solved by single-crystal X-ray diffraction (SC-XRD), revealing that MNP$_3$ adopts the polar trigonal *R3* space group at room temperature (Table S2). The purity of the crystal can be judged from the good agreement between experimental and simulated powder X-ray diffraction (PXRD) patterns (Fig. S1). The overall structure is isostructural to MDABCO-NH$_4$-I$_3$ (MNI$_3$), but with a bigger unit cell ($a = b = c = 7.844$ Å, $\alpha = \beta = \gamma = 84.857°$ for MNP$_3$ and $a = b = c = 7.259$ Å, $\alpha = \beta = \gamma = 84.767°$ for MNI$_3$) (Figs. S2, S3). The synthesized crystals are rhombohedral shaped and are ⟨100⟩ textured according to XRD (Figs. S4 and S5). There are multiple hydrogen bonds in MNP$_3$, including N–H⋯F and C–H⋯F, with bond lengths ranging from 2.2 to 4.0 Å (Figs. S6 and S7). Particularly,

**(a)**

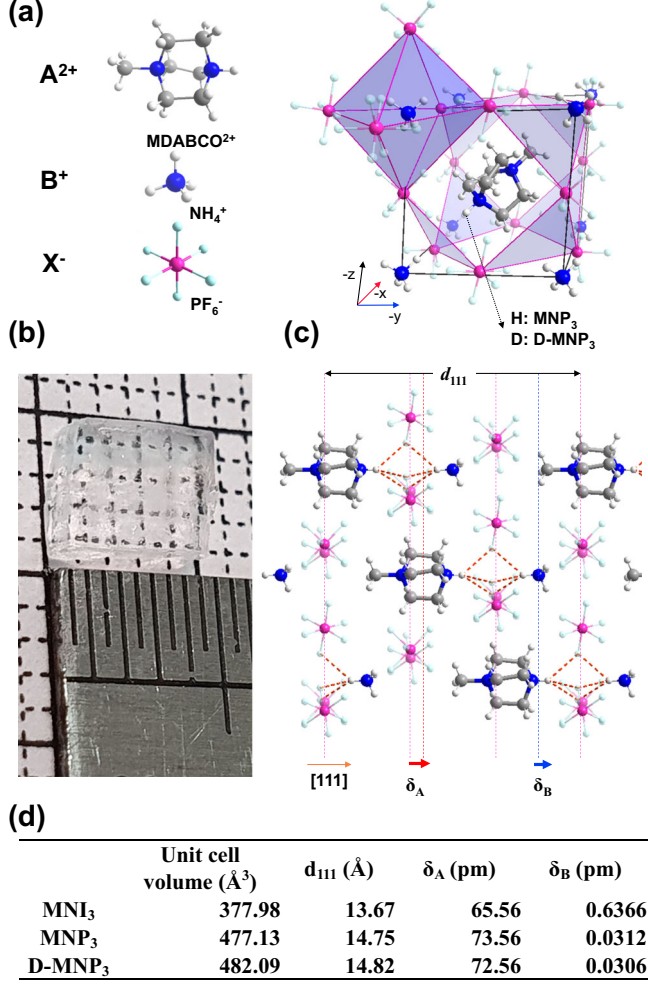

**(b)**

**(c)**

**(d)**

| | Unit cell volume ($\text{Å}^3$) | $d_{111}$ (Å) | $\delta_A$ (pm) | $\delta_B$ (pm) |
|---|---|---|---|---|
| **MNI$_3$** | 377.98 | 13.67 | 65.56 | 0.6366 |
| **MNP$_3$** | 477.13 | 14.75 | 73.56 | 0.0312 |
| **D-MNP$_3$** | 482.09 | 14.82 | 72.56 | 0.0306 |

**Fig. 2 Crystal structure of MDABCO-NH$_4$-(PF$_6$)$_3$ and the polar displacement direction. a** The A$^{2+}$, B$^+$ and X$^-$ components, and their relative positions in MDABCO-NH$_4$-(PF$_6$) unit cell; **b** as-grown 5 × 4 × 1 mm (l × d × h) size single-crystal. **c** Arrangement of different ions along the [111] polar axis, with arrows indicating the displacement of MDABCO molecules. The N–H⋯F hydrogen bonds ($d < 2.4$ Å) are highlighted with dotted orange lines. **d** Table of unit cell volume, distance in [111] direction ($d_{111}$), displacement of A$^{2+}$ cation ($\delta_A$), and B$^+$ cation ($\delta_B$) for MNI$_3$, MNP$_3$ and D-MNP$_3$.

the N–H bond of MDABCO is aligned in the [111] polarization direction of MNP$_3$, and form hydrogen bonds with the three nearest fluorine atoms (Fig. 2c). The strong hydrogen bond is revealed from the bond distance, the N–H⋯F hydrogen bonds (2.4 Å) are much shorter than N–H⋯I in MNI$_3$ (3.2 Å). The strong N–H⋯F bond causes MDABCO to be pulled towards [111] direction. As a result, the displacement of MDABCO in the [111] direction is 73.56 pm (4.98%) for MNP$_3$, which is longer than MNI$_3$ whose displacement is 65.56 pm (4.79%) (Fig. 2d).

To confirm the effect of hydrogen bond on ferroelectricity, we also synthesized deuterated MNP$_3$ (D-MNP$_3$) in which the proton in N–H bond of MDABCO is substituted by deuterium[19–21]. SC-XRD shows that unit cell dimension of D-MNP$_3$, with $a = b = c = 7.872$ Å, and $\alpha = \beta = \gamma = 84.780°$, is slightly larger and shows less octahedral tilt compared to MNP$_3$. This increased unit cell is apparent in the PXRD (Fig. S8). The unit cell dimension ($d_{111}$) stretches along the hydrogen-bond direction, i.e. [111] direction, which manifests the Ubbelohde effect[22]. This phenomenon is caused by the longer N–D bond than N–H, so that

displacement of MDABCO ($\delta_A$) is reduced from 73.56 pm to 72.56 pm by deuteration (Fig. 2d). The longer N–D bond is caused by the stronger D-F bonding, which reduces the electron density on D; the distance between the two nitrogen atoms in MDABCO is increased from 2.501 to 2.518 Å by deuteration.

A hallmark of ferroelectricity is the ferroelectric-to-paraelectric phase transition. Phase transition can be detected using differential scanning calorimetry (DSC) measurement, where a sharp endothermic peak can be seen at 311 K (Fig. 3a). MNP$_3$ has a larger cuboctahedral unit cell than MNI$_3$, thus MDABCO in MNP$_3$ can rotate more freely, resulting in a lower phase transition temperature than MNI$_3$ (448 K). The correlation between free space rotation and phase transition temperature is also demonstrated by the isostructural perovskite MDAB-Rb-I$_3$, where its lower phase transition temperature (430 K) than MNI$_3$ is correlated to its larger unit cell[23].

The phase transition temperature of D-MNP$_3$ is 320 K, which is 19 K higher than MNP$_3$. The effect of isotope substitution on phase transition temperature is a characteristic feature of hydrogen-bonded ferroelectrics such as KDP family[19,24], glycine phosfite[20], and supramolecular complexes[4,25,26]. This implies that all-organic perovskite share characteristics of organic ferroelectrics although the origin of its polarization arises from ionic displacement like inorganic perovskites. The higher phase transition temperature in deuterated perovskite originates from its stronger hydrogen bonds. A stronger hydrogen bond increases the barrier for the rotation of MDABCO in the PE phase similar to H/F substituted molecular ferroelectrics[27]. In addition, the much larger enthalpy change of D-MNP$_3$ (94.5 J g$^{-1}$) as compared with MNP$_3$ (30.4 J g$^{-1}$) is indicative of the stronger hydrogen bonding in D-MNP$_3$ as compared to MNP$_3$.

Ferroelectric-paraelectric phase transition was confirmed by a sharp change in second harmonic generation (SHG) intensity across the Curie temperature. The SHG signal originates from the non-centrosymmetric structure of materials, so it vanishes when centrosymmetry is restored in the paraelectric phase. Figure 3b shows the temperature-dependent SHG signal of MNP$_3$ and D-MNP$_3$ collected in the temperature range from 305 to 340 K. The SHG intensity of MNP$_3$ gradually decreases with temperature and vanishes around 315 K due to the recovery of centrosymmetry in the paraelectric phase. Similarly, the SHG signal of D-MNP$_3$ vanishes around 323 K when the crystal transits to the centrosymmetric phase, in line with the DSC results.

MNP$_3$ and D-MNP$_3$ undergo a phase transition from the ferroelectric $R3$ space group to the paraelectric cubic $P432$ space group similar to what has been reported for isostructural perovskite structures[14,23]. The paraelectric unit cell dimensions of MNP$_3$ are $a = b = c = 7.9557$ Å, and $\alpha = \beta = \gamma = 90°$, and for paraelectric D-MNP$_3$, $a = b = c = 7.9154$ Å, and $\alpha = \beta = \gamma = 90°$ which confirmed by SC-XRD. To monitor structural changes during the phase transition process, temperature-dependent powder X-ray diffraction (PXRD) for both MNP$_3$ and D-MNP$_3$ were recorded. At the transition temperature of each sample, the XRD peaks of (100), (111) and (200) at 11.26°, 20.14° and 22.68° (Fig. S9) shift to lower angles of 11.12°, 19.36° and 22.4°, respectively, indicating that the unit cell dimension has expanded; the vanishing of other peaks indicate that a more symmetric cubic cell is achieved in paraelectric phase. It is observed that the (111) peaks due to ferroelectric phase at 20.14° and the paraelectric phase at 19.36° coexist without other intermediate states during phase transition (Fig. 3c 313 K and 3d 323 K), this suggests that the phase change is a discontinuous order-disorder type (first-order)[28].

The fingerprints of ferroelectric-to-paraelectric phase transition can also be observed in the solid-state $^1$H NMR spectrum (Fig. 3e). The CH$_2$ and CH$_3$ peaks of MDABCO at 3.9 and 3.2

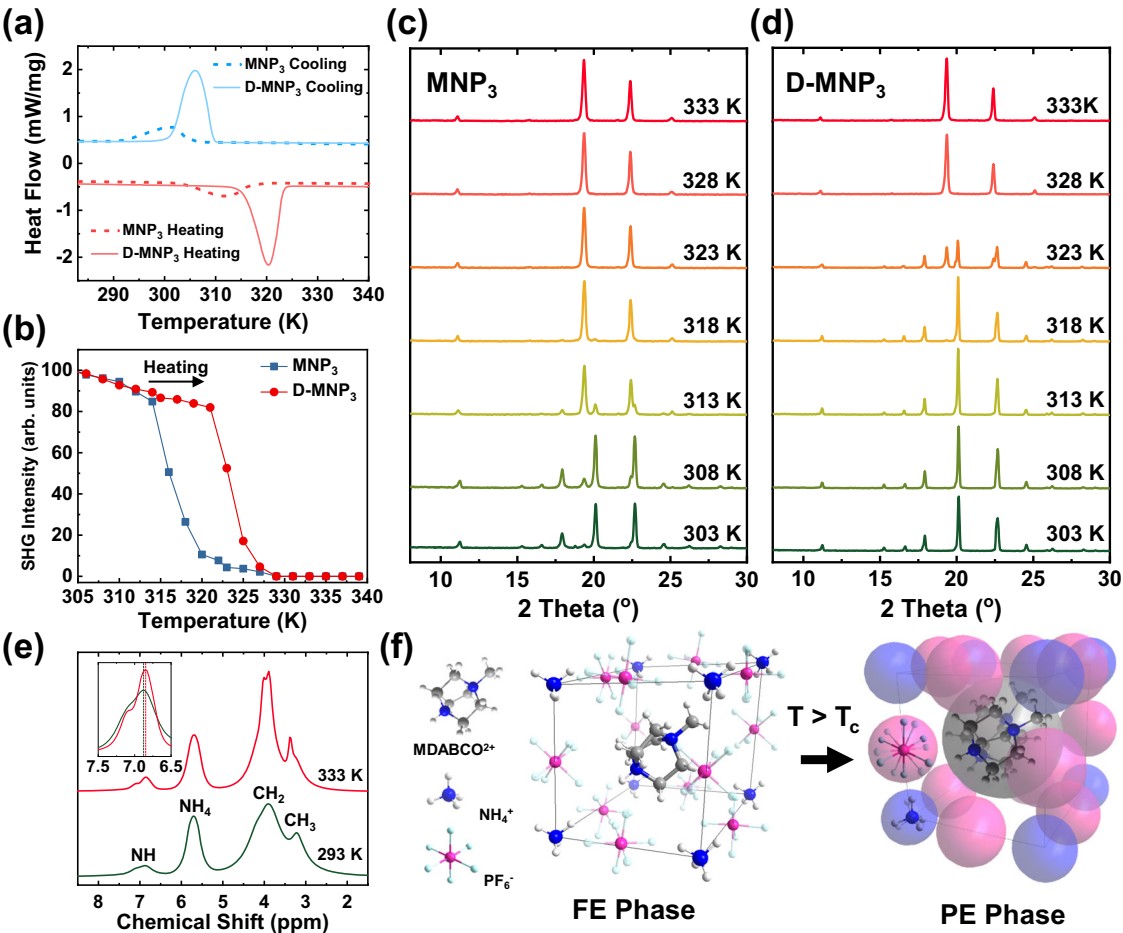

**Fig. 3 The ferroelectric-to-paraelectric transition of MNP₃ and D-MNP₃ crystal. a** DSC and **b** temperature-dependent SHG intensity change of MNP₃ and D-MNP₃ revealing phase transition from ferroelectric (FE) phase to paraelectric (PE) phase, and that D-MNP₃ has a higher $T_c$ than MNP₃. Temperature-dependent PXRD result of MNP₃ (**c**) and D-MNP₃ (**d**). **e** Solid-state ¹H NMR spectrum of MNP₃ at 293 K (FE phase) and 333 K (PE phase). Inset shows peak shift (6.88–6.84) of N–H in MDABCO indicating shrinkage of N–H bond. **f** Schematic illustration of MNP₃ structure in ferroelectric (FE) phase and paraelectric (PE) phase. In the PE phase, molecules rotate freely.

ppm become sharper in the paraelectric phase due to the free rotation of the MDABCO molecule. The peak of N–H of MDABCO at 6.88 is split into two due to a second-order effect of dipolar coupling to the quadrupolar nitrogen-14 nuclei[29]. This peak shift from 6.88 to 6.84 ppm that accompanies phase transition is indicative of a reduction of the N–H bond length because of the weakening of the hydrogen bonds in the paraelectric phase[30]. Furthermore, the P–F distance is also shortened in the paraelectric phase at 333 K as reflected from the solid-state NMR, where the J-coupling constant is reduced (Fig. S10). Therefore, solid-state NMR results validate that N–H and P–F bonds are elongated in the ferroelectric phase due to strong hydrogen bonds, whereas in the PE phase, both MDABCO and PF₆ rotate freely and N–H and P–F bonds are shorter. The orientational disorder of molecules in the paralelectric phase has been extensively studied previously by Jun Harada and others[31–35]. One characteristic of this rotation disorder is that under polarized light microscopy, the iridescence of the ferroelectric phase due to birefringence vanishes upon changing to the paraelectric phase, which is clearly confirmed for our MNP₃ crystal (supplementary information Fig. S11).

Temperature-dependent dielectric permittivity was also used to detect the phase transition of the crystals as a ferroelectric-to paraelectric transition will result in an anomaly in the dielectric permittivity. Both MNP₃ and D-MNP₃ show an abrupt increase

of dielectric permittivity around their phase transition temperature (Fig. S12 a and b). We also fitted the inverse of dielectric constant with the temperature at 100 kHz (Fig. S12 c and d) by Curie-Weiss law that is, $\varepsilon = C/(T - T_0)$ where $\varepsilon$ is the dielectric constant, C is the Curie constant, T is the temperature, and $T_0$ is the Curie temperature. Linear fitting of $1/\varepsilon$ with the temperature at 100 kHz allows the Curie temperatures of MNP₃ and D-MNP₃ to be determined as 307 K and 316 K, and the Curie constants as 254 K and 382 K, respectively. The dielectric permittivity around the phase transition temperature is strongly dependent on frequency. A smaller dielectric permittivity at a higher frequency means the ferroelectric-to-paraelectric transition is the order-disorder type[36]. D-MNP₃ shows a much higher dielectric permittivity than MNP₃, which may be explained by the higher polarization of the former arising from its stronger H-bonding[37].

The polarization (P) versus electric field (E) measurement as shown in Fig. 4a exhibits a hysteresis loop characteristic of a ferroelectric crystal. The remnant polarization $P_r$ is 5.7 μC/cm², and the coercive field $E_c$ ranges from 54 to 110 kV/cm depending on the applied frequency. The coercive field of MNP₃ is much higher than isostructural organic perovskite MNI₃ (12 kV/cm), this is in line with the expectation that a stronger hydrogen bond leads to a higher coercive field. Compared with the relatively low coercive field of BTO (10 kV/cm) and PZT (76 kV/cm)[38], and the exceedingly large field of 500 kV/cm for PVDF[39], MNP₃ exhibits

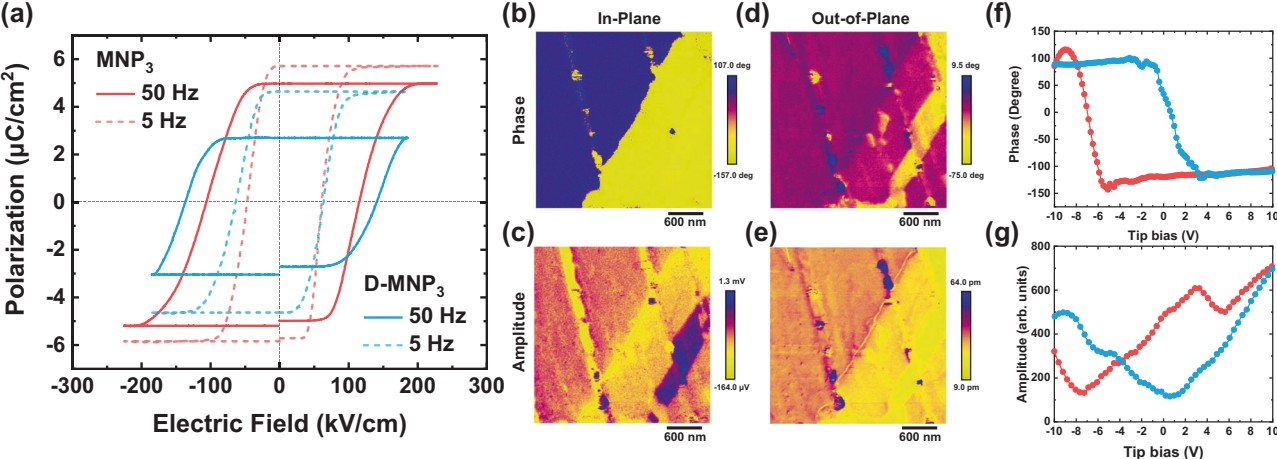

**Fig. 4 Polarization-electric field (*P-E*) loop and ferroelectric domains. a** *P-E* loop of MNP₃ and D-MNP₃ at 50, and 5 Hz. PFM phase images of **b** in-plane and **d** out-of-plane polarization. PFM amplitude images of **c** in-plane and **e** out-of-plane polarization. PFM phase (**f**) and amplitude (**g**) switching curve from MNP₃ thin film.

a mid-range coercive field that translates to a threshold voltage of 1 V for a 90 nm thick film, which is compatible with the standard operating condition of thin-film ferroelectric random-access memory (FeRAM) devices. Furthermore, the coercive field of D-MNP₃ is increased to 138 kV/cm at 50 Hz, this agrees with the fact that D-MNP₃ has stronger hydrogen bonds. These results prove that we can control the coercive field of the ferroelectric materials by tuning the intermolecular hydrogen-bond strength. The smaller polarization value of D-MNP₃ (4.6 μC/cm²) than MNP₃ (5.7 μC/cm²) at 5 Hz can be attributed to two factors: first, D-MNP₃ (482.09 Å³) has a larger unit cell volume than MNP₃ (477.13 Å³); secondly, the A-site cation displacement of D-MNP₃ (72.56 pm) is shorter than MNP₃ (73.56 pm) due to the Ubbelohde effect. The polarization values at 5 Hz differ by about 23%, but it increases to 84% at 50 Hz where the polarization value of D-MNP₃ is 2.7 μC/cm² and MNP₃ is 5.0 μC/cm². Considering that the decline in polarization at high frequency is due to the inability of the switching dipoles to keep up with the oscillating field, this effect is more apparent in D-MNP₃ due to its stronger hydrogen bond. (Fig. S13).

According to the $432F3$ transition, MNP₃ has four polar axis and eight polarization directions[40]. Since the polar axes are in between the in-plane ($y$-axis) and out-of-plane ($z$-axis) directions, ferroelectric domains whose polarization directions are not perpendicular to these can be observed in both in-plane and out-of-plane phase and amplitude images using piezoresponse force microscopy (PFM). In Fig. 4b, a 180° in-plane domain change can be seen clearly, which indicates two polarized domains in the $y$-axis direction (+y and −y). In the amplitude scan of the in-plane PFM image (Fig. 4c), there is a region that overlaps with some parts of the out-of-plane phase (Fig. 4d), which is indicative of canted polarizations with vectors in both −y and −z directions (Fig. S14 b green area). We could also identify areas (red in Fig. S14) with canted polarizations and vector components in −y and −z direction, and another two regions with canted polarizations and vector components along +y and −y region according to the in-plane phase image. Therefore, from the PFM images, we can observe four distinct polarization domains (Fig. S14 b and c), similar to the previously reported isostructural MNI₃[14]. Due to the high coercive field of MNP₃, it was difficult to perform phase switching on a thick, single-crystal sample using PFM. Instead, ferroelectric switching was performed on a 1 μm-thick thin film of MNP₃ prepared by a spray-coating method (Fig. S15). A clear 180-degree phase reversal (Fig. 4f) and butterfly curve (Fig. 4g)

were observed by sweeping the tip voltage from −10 to 10 V, indicating that ferroelectric domains can be electrically switched.

To validate our experimentally measured polarization values and coercive field, we performed density functional theory (DFT) calculations of both MNI₃ and MNP₃. First of all, spontaneous polarization value was calculated by Berry phase calculation developed by King-Smith and Vanderbilt[41,42]. We constructed a $\sqrt{2} \times \sqrt{2} \times 1$ supercell based on the unit cell of the ferroelectric phase, and built the centrosymmetric reference phase via rotation, displacement and distortion of the components, after which two MDABCO molecules in the simulation cell are aligned anti-parallel to each other (Fig. 5a). The calculated polarization of MNP₃ is shown in Fig. 5b, where it varies continuously from 0 to 6.5 μC/cm² along the dynamic path, thus the spontaneous polarization agrees well with the experimental value of 5.7 μC/cm². Although the larger displacement of MDABCO in MNP₃ than MNI₃ (Fig. 2d) gives a larger dipole moment in the former, the larger unit cell volume of MNP₃ than MNI₃ offsets the increased polarization as the overall polarization is defined by the sum of dipole moment per unit volume. Furthermore, in MNI₃, there is a large polarization contribution from the off-centre displacement of NH₄⁺ at B site ($\delta_B = 0.6366$ pm) because of its larger $r_B/r_X$ ratio (0.664) compared to the typical octahedron ratio (0.414–0.592) predicted by Pauling's rule. On the other hand, the NH₄⁺ ion in MNP₃ is located at the centre of the octahedron ($\delta_B = 0.0312$ pm) with a $r_B/r_X$ ratio of 0.570, thus there is no off-centre displacement.

The displacement of MDABCO in the polarization direction with respect to the MNP₃ framework is the primary driver of the ferroelectricity of MNP₃. The large electrostatic potential differences between MDABCO-NH and PF₆ increase the hydrogen-bond strength compared to MNI₃. Mulliken charge calculations for MNP₃ and MNI₃ reveal that PF₆⁻ has a more pronounced inductive effect than I⁻ on the H atom in MDABCO (Fig. 5c), thus the stronger N–H⋯F hydrogen retards the polarity reversal process of organic ferroelectric perovskites. Therefore, any reinforcement of hydrogen bonding will hinder the ferroelectric switching process, since multiple hydrogen bonds need to be successively broken and established during the rotation of MDABCO, as illustrated in Fig. 5a. This is the root cause of the enhanced coercive field for MNP₃.

In conclusion, incorporating stronger hydrogen bonds in MDABCO-NH₄-I₃ by the substitution of PF₆⁻ for I⁻ increase the coercive field by an order of magnitude from ~12 kV/cm to

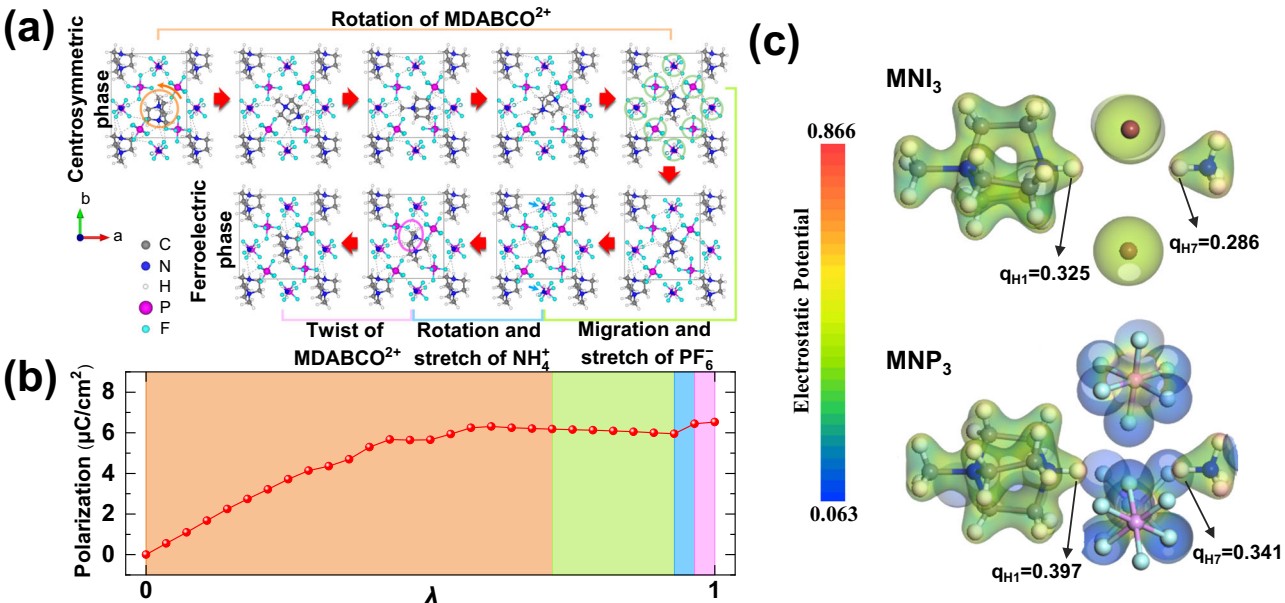

**Fig. 5 Berry phase calculation of spontaneous polarization in MNP₃. a** The dynamic path of ferroelectric phase transition in $MNP_3$. **b** The polarization value of $MNP_3$ was obtained by Berry phase calculations. The path connects the centrosymmetric reference phase ($\lambda = 0$) to the ferroelectric phase ($\lambda = 1$). **c** Electrostatic potential of $MNI_3$ and $MNP_3$, and the Mulliken charges of H atoms participating in hydrogen bonds.

110 kV/cm. This enables a threshold voltage of 1 V for a 90 nm thick film, which complies with the miniaturization requirement of FeRAM devices. Both experimental results and DFT calculations provide robust evidence for the mechanism of MDABCO rotation upon ferroelectric switching and phase transition, the dynamics of which are modulated by the intermolecular hydrogen bonding interactions between $MDABCO^{2+}$ and $PF_6^-$. This work demonstrates that regulating intermolecular interactions between the cation and anion can be used for the engineering of the coercive field in organic ferroelectric materials.

## Methods

**Synthesis.** Detailed synthetic methods for preparing precursors are provided in Supplementary Information. For growing $MNP_3$ crystal, 209 mg (0.5 mmol) of $MDABCO-(PF_6)_2$ and 81.5 mg (0.5 mmol) of $NH_4PF_6$ were dissolved in acetonitrile until saturated, and the parallelogrammic single-crystals were grown after slow evaporation.

**Characterization.** All chemicals were purchased from Sigma–Aldrich without further purification. $^1H$ and $^{13}C$ nuclear magnetic resonance (NMR) was taken by AVII 400 MHz NMR spectrometer of Bruker. Thermogravimetric analyses (TGA) were performed under a nitrogen atmosphere with a heating rate of 10 °C/min using a TA Instruments Trios V3.1 thermogravimetric analyzer. Differential scanning calorimetry (DSC) scans were performed under a nitrogen atmosphere with a heating rate of 10 °C/min using Mettler-Toledo DSC. The dielectric constant was measured by CVU unit in Keithley-SCS4200 with the pelleted sample. Powder X-ray diffraction (PXRD) patterns were recorded on a Bruker D8 Focus Powder X-ray diffractometer using Cu Kα radiation (40 kV, 40 mA) at room temperature. Ferroelectric $P$-$E$ curve was measured with Precision Multiferroic II Ferroelectric Test System of the Radiant Technologies with high voltage amplifier. Piezoresponse force microscopy (PFM) tests were performed on Bruker Dimension Icon Atomic Force Microscope with grown crystals or spray-coated samples on ITO.

**Density functional theory (DFT) calculations.** We performed the Berry phase calculations[41,42] within the DFT framework as implemented in the Vienna ab initio simulation package (VASP)[43,44]. The exchange-correlation interactions were treated within the Perdew-Burke-Ernzerh (PBE) generalized gradient approximation[45]. To complement the deficiencies of DFT in treating dispersion interactions, the third-generation (D3) van der Waals corrections proposed by Grimme[46] were employed. The plane-wave cutoff energy was set to 520 eV, and the $k$-point mesh was set to $3 \times 3 \times 4$. The polarization was calculated using a supercell twice the size of the unit cell so that a centrosymmetric reference phase can be constructed. A convergence threshold of 0.01 eV/Å in force was reached in structural optimization. Electrostatic potential and Mulliken charge were calculated by DMol3[47,48] code in Materials Studio using a double numerical polarized basis set and PBE[45] exchange-correlation functional.

**Reporting summary.** Further information on research design is available in the Nature Research Reporting Summary linked to this article.

## Data availability

All data generated and analyzed in this study are included in the Article and its Supplementary Information, and are also available from corresponding authors upon request. Crystallographic data for this paper can be obtained free of charge from the Cambridge Crystallographic Data Centre via www.ccdc.cam.ac.uk/data_request/cif. CCDC- 2085249 ($MNP_3$ at RT) and CCDC- 2085250 (D-$MNP_3$ at RT).

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

## Acknowledgements

K.P.L. would like to acknowledge the Singapore's Ministry of Education Tier 2 grant (MOE2019-T2-1-037), the Shenzhen Peacock Plan (Grant No. KQTD2016053112042971). F.P. would like to acknowledge the Guangdong Special Support Program, and Soft Science Research Project of Guangdong Province (No. 2017B030301013). K.Y. would like to acknowledge partial supports by A*STAR, Singapore, RIE2020 Advanced Manufacturing and Engineering (AME) Programmatic Fund, (Grant No. A20G9b0135).

## Author contributions

H.S.C. designed, synthesized and characterized the MNP3 crystals. K.P.L. supervised the research. S.L., S.Z. and F.P. did DFT calculations. I.H.P. did a single-crystal XRD structure analysis. W.H.L. and K.Y. did a ferroelectric experiment and analysis. Z.Z. and Q.H.X. did a temperature-dependent SHG experiment. K.C.K. did PFM measurement. L.W. did temperature-dependent dielectric measurement. I.H.O. did an analysis on the deuteration effect. H.S.C., S.L. and K.P.L. wrote the manuscript in consultation with all authors. H.S.C. and S.L. contributed equally.

## Competing interests

The authors declare no competing interests.
