## [Peer review file · Nature Communications]

REVIEWER COMMENTS

Reviewer #1 (Remarks to the Author):

In this work, it is shown that the coercive field of all-organic ferroelectric materials can be changed by tuning the strength of non-covalent interactions. The earlier reported all-organic ferroelectric crystals, MDABCO-NH₄-I3 showed a small coercive field of about 12 kV/cm while a slight modification of the composition to MDABCO-NH₄-(PF₆)₃, by introducing PF₆ ions, the authors show that the coercive field is increased by an order of magnitude. This high E_c , which is attributed to the strengthened hydrogen bonding interactions, is important in the context of practical applications. The work is well done and the article is easy to follow. The concept presented here has general implications to the design of ferroelectric materials for use in practical applications. However, there are some minor issues which needs to be addressed before the article is considered for publication.

1. From the manuscript, it is not clear how the authors identified the crystal phases. Ideally, one shall present the face indexing data from single crystal XRD. This needs to be added so that the axes in structure with respect to the crystal in PFM and other studies are clear and accurate. Phase indexing is required to visually understand the polarisation direction from the crystal point of view.

2. Authors suggest, using the deuterated variant, that the strengthening of hydrogen bonds increases the E_c as desired for applications. Although this is a good supportive evidence, other factors, such as asphericity, globularity and symmetry of the ions can't be ruled out. For instance, the PF₆ ions are very large and possess a different symmetry. In plastic crystal (orientationally disordered) phases, it is well known that the molecular rotations depend on many factors. It is essential that the authors comment on possible contributions from other factors as well. See the examples discussed in this review, Chem. Soc. Rev., 2020, 49, 8878-8896 and the references therein.

3. Activation of molecular rotations upon phase transformation at higher temperature can be further characterized by a simple experiment, i.e. the host-stage microscopy under cross-polarizer. Upon activation of rotations, the iridescence of the crystals will vanish. Authors may attempt this and present the evidence in supporting information. Upon activation of rotations, the crystal symmetry increases, which is evident from the PXRD data (decreased number of peaks).

4. The text in the manuscript and the figures are not in sink. Authors should pay more attention to this aspect and revise it suitably.

5. There are some typographical errors. One round of proof reading is needed.

6. Page 5: "single crystal of 5 mm dimension could be grown". Dimension for 3d objects should have at least two lengths, if not three. Please provide the dimensions properly.

7. In PE phase molecules rotate freely. Such crystals are known as "plastic crystals" or "orientationally disordered phases". Some representative articles should be cited from this topic so that the readers from different backgrounds can benefit. There are some good articles by Jun Harada, C M Reddy and others.

Reviewer #2 (Remarks to the Author):

Choi et al. report the tailoring the coercive field in a newly synthesized metal-free perovskite. The ferroelectric-to-paraelectric phase transition temperature and coercive field are found to be regulated by hydrogen bonds. It is of great significance to search or rational design molecular ferroelectric

crystals with suitable coercive field, which can not only provide better understanding of various origins of ferroelectricity, but also provide a good material template for the application. This work enables a threshold voltage of 1 V for a 90 nm thick film, which is an important milestone in the development of molecular ferroelectrics. Therefore, I recommend that this manuscript be published in the current version.

Reviewer #3 (Remarks to the Author):

Metal-free perovskite ferroelectrics have attracted intense attention because they are complementary ferroelectric materials of oxides due to their molecular designability, mechanical flexibility and ease of synthesis. In this work, the authors report a much improved coercive field of a metal-free perovskite ferroelectric, [MDABCO][NH₄](PF₆)₃, by reinforcing the hydrogen bonding. Notably, the coercive field of [MDABCO][NH₄](PF₆)₃ is an order of magnitude higher (110 kV/cm) than that of [MDABCO][NH₄]I₃ (12 kV/cm), which is promising for the miniaturization of ferroelectric devices composed of molecular ferroelectrics. This study unambiguously highlights the wide chemical tunability of ferroelectric properties in these molecular systems, which are inaccessible in their conventional inorganic counterparts. All the characterizations are comprehensive and the results are convincing. In my opinion, these findings are very significant for advancing the field of molecular ferroelectric materials, which will attract broad interests and diverse readership of Nature Communications. Therefore, I recommend publishing this manuscript after a minor revision. The authors can consider the following comments and suggestions to improve their manuscript:

1. I do not fully agree with the usage of the terminology of "all organic perovskites". PF₆⁻ is certainly an inorganic ion, so [MDABCO][NH₄](PF₆)₃ is not all organic. I suggest the authors change "all organic perovskites" to "metal-free perovskites" to be consistent with the original paper published by Xiong et al. in 2018 (Science 2018, 361, 151). In addition, the authors only did one material (not considering the deuterated one here), so I feel the title presents is a bit grander story than the paper delivers. While I'm not disputing the conclusion here can be extended to many other systems, it would be reasonable to give a specific title.
2. In Fig. 3b, the SHG plots need to be mentioned whether they are from cooling or heating. In addition, the text of the perpendicular axis needs to be changed to "SHG intensity". The font of texts of both x- and y- axes in Figs. 3b and 3e are not consistent with that in others. In Fig. 3e, I wonder why the authors did not measure the deuterated sample instead (which can give much better signal)? In Fig. 3f, the atom codes need to be provided (same as Fig. 2).
3. In Fig 4a, why the deuterated sample show significantly reduced polarization compared with the original phase at the same conditions? Any possible reason? The PFM Figs. 4f and 4g do not look perfect, can the authors give some explanation?
4. Some format problems: the format of references need to be reorganized to make them consistent: refs 2-5, 10, 12 etc.; the letters in cell parameters and space groups need to be italic etc.
5. Some important references need to be added: Science 311 (2006) 1270; Nature Materials Reviews, 2017, 2, 16099; APL Materials, 2021, 9, 070401; Wiley book: <https://onlinelibrary.wiley.com/doi/book/10.1002/9783527344338>.

Point by Point response for

Tailoring the Coercive Field in Ferroelectric **Metal-Free** Perovskites by Hydrogen Bonding

REVIEWER COMMENTS

Reviewer #1 (Remarks to the Author):

In this work, it is shown that the coercive field of all-organic ferroelectric materials can be changed by tuning the strength of non-covalent interactions. The earlier reported all-organic ferroelectric crystals, MDABCO-NH₄-I₃ showed a small coercive field of about 12 kV/cm while a slight modification of the composition to MDABCO-NH₄-(PF₆)₃, by introducing PF₆ ions, the authors show that the coercive field is increased by an order of magnitude. This high E_c, which is attributed to the strengthened hydrogen bonding interactions, is important in the context of practical applications. The work is well done and the article is easy to follow. The concept presented here has general implications to the design of ferroelectric materials for use in practical applications. However, there are some minor issues which needs to be addressed before the article is considered for publication.

1. From the manuscript, it is not clear how the authors identified the crystal phases. Ideally, one shall present the face indexing data from single crystal XRD. This needs to be added so that the axes in structure with respect to the crystal in PFM and other studies are clear and accurate. Phase indexing is required to visually understand the polarisation direction from the crystal point of view.

Answer: We have added an optical image of crystal (Supplementary information Figure S4) showing a rhombohedral crystal shape that shows the <100> texture. The face of the single crystal was confirmed by XRD that shows only diffractions from (100), (200), (300), and (400) planes (Supplementary information Figure S5). The related sentence is added on manuscript page 7, lines 107-108.

2. Authors suggest, using the deuterated variant, that the strengthening of hydrogen bonds increases the E_c as desired for applications. Although this is a good supportive evidence, other factors, such as asphericity, globularity and symmetry of the ions can't be ruled out. For instance, the PF_6 ions are very large and possess a different symmetry. In plastic crystal (orientationally disordered) phases, it is well known that the molecular rotations depend on many factors. It is essential that the authors comment on possible contributions from other factors as well. See the examples discussed in this review, Chem. Soc. Rev., 2020, 49, 8878-8896 and the references therein.

Answer: Thanks for the nice suggestion. However, by comparing the isostructural series of perovskites containing MDABCO as A site cation (Supplementary Information Table S1), we can conclude that high coercive field (E_c) of MNP_3 is strongly correlated to the hydrogen bond, while the phase transition temperature (T_c) is strongly correlated to Goldschmidt tolerance factor.

First of all, even though we change spherical ions to molecular ions (Rb to NH_4 or I to PF_6), crystal symmetry remains same in both low temperature phase ($R3$) and high temperature phase ($P432$). So symmetry change in components doesn't affect the crystal structure and whole symmetry.

Secondly, the coercive field of MDABCO- NH_4 - I_3 is smaller than MDABCO-Rb- I_3 due to larger cuboctahedral space which is evident in larger Goldschmidt tolerance factor (0.935 for MDABCO- NH_4 - I_3 and 0.927 for MDABCO-Rb- I_3). However, although MDABCO- NH_4 - $(PF_6)_3$ has the *largest cuboctahedral space* among the three with its Goldschmidt tolerance factor of 0.914, its coercive field is the *largest* among them. Therefore, we infer that strong hydrogen bond is the main reason for high coercive field of MDABCO- NH_4 - $(PF_6)_3$.

Lastly, phase transition temperature is strongly correlated to the Goldschmidt tolerance factor because that concerns the size of the cuboctahedral space for MDABCO molecule to be rotated freely. The larger the cuboctahedral space, the easier for MDABCO molecule to be rotated. Accordingly, the transition temperatures of MDABCO- NH_4 - I_3 , MDABCO-Rb- I_3 , and MDABCO- NH_4 - $(PF_6)_3$ are 448, 430, and 311 K, respectively, and they follow the sequence of their Goldschmidt tolerance factor 0.935, 0.927, and 0.914, where a smaller Goldschmidt tolerance factor means a larger cuboctahedral space.

3. Activation of molecular rotations upon phase transformation at higher temperature can be further characterized by a simple experiment, i.e. the host-stage microscopy under cross-polarizer. Upon activation of rotations, the iridescence of the crystals will vanish. Authors may attempt this and present the evidence in supporting information. Upon activation of rotations, the crystal symmetry increases, which is evident from the PXRD data (decreased number of peaks).

Answer: We have added polarized light microscopy images of single crystal MNP_3 during ferroelectric to paraelectric phase change in Supplementary Information Figure S11 and provided the explanation in the revised manuscript page 11 line 191-195. It is clearly seen that the iridescence of the crystals vanished upon phase transition.

4. The text in the manuscript and the figures are not in sink. Authors should pay more attention to this aspect and revise it suitably.

Answer: Thanks for bringing this to our attention, we have rearranged the supplementary figure and corrected the manuscript accordingly.

5. There are some typographical errors. One round of proof reading is needed.

Answer: Thanks, we have corrected the typographical errors.

6. Page 5: "single crystal of 5 mm dimension could be grown". Dimension for 3d objects should have at least two lengths, if not three. Please provide the dimensions properly.

Answer: We have changed from 5 mm dimension to $5 \times 4 \times 1$ mm ($l \times d \times h$) dimension in revised manuscript page 5 line 89.

7. In PE phase molecules rotate freely. Such crystals are known as "plastic crystals" or "orientationally disordered phases". Some representative articles should be cited from this topic so that the readers from different backgrounds can benefit. There are some good articles by Jun Harada, C M Reddy and others.

Answer: We have added some comments and five related references (including those by Jun Harada, C M Reddy and others) in our revised manuscript page 11 line 190-191.

Reviewer #2 (Remarks to the Author):

Choi et al. report the tailoring the coercive field in a newly synthesized metal-free perovskite. The ferroelectric-to-paraelectric phase transition temperature and coercive field are found to be regulated by hydrogen bonds. It is of great significance to search or rational design molecular ferroelectric crystals with suitable coercive field, which can not only provide better understanding of various origins of ferroelectricity, but also provide a good material template for the application. This work enables a threshold voltage of 1 V for a 90 nm thick film, which is an important milestone in the development of molecular ferroelectrics. Therefore, I recommend that this manuscript be published in the current version.

Answer: Thank u very much

Reviewer #3 (Remarks to the Author):

Metal-free perovskite ferroelectrics have attracted intense attention because they are complementary ferroelectric materials of oxides due to their molecular designability, mechanical flexibility and ease of synthesis. In this work, the authors report a much improved coercive field of a metal-free perovskite ferroelectric, [MDABCO][NH₄](PF₆)₃, by reinforcing the hydrogen bonding. Notably, the coercive field of [MDABCO][NH₄](PF₆)₃ is an order of magnitude higher (110 kV/cm) than that of [MDABCO][NH₄]I₃ (12 kV/cm), which is promising for the miniaturization of ferroelectric devices composed of molecular ferroelectrics. This study unambiguously highlights the wide chemical tunability of ferroelectric properties in these molecular systems, which are inaccessible in their conventional inorganic counterparts. All the characterizations are comprehensive and the results are convincing. In my opinion, these findings are very significant for advancing the field of molecular ferroelectric

materials, which will attract broad interests and diverse readership of Nature Communications. Therefore, I recommend publishing this manuscript after a minor revision. The authors can consider the following comments and suggestions to improve their manuscript:

1. I do not fully agree with the usage of the terminology of “all organic perovskites”. PF_6^- is certainly an inorganic ion, so $[\text{MDABCO}][\text{NH}_4](\text{PF}_6)_3$ is not all organic. I suggest the authors change “all organic perovskites” to “metal-free perovskites” to be consistent with the original paper published by Xiong et al. in 2018 (Science 2018, 361, 151). In addition, the authors only did one material (not considering the deuterated one here), so I feel the title presents is a bit grander story than the paper delivers. While I’m not disputing the conclusion here can be extended to many other systems, it would be reasonable to give a specific title.

Answer: We have changed the word “all-organic” to “metal-free” in title of the revised manuscript.

2. In Fig. 3b, the SHG plots need to be mentioned whether they are from cooling or heating. In addition, the text of the perpendicular axis needs to be changed to “SHG intensity”. The font of texts of both x- and y- axes in Figs. 3b and 3e are not consistent with that in others. In Fig. 3e, I wonder why the authors did not measure the deuterated sample instead (which can give much better signal)? In Fig. 3f, the atom codes need to be provided (same as Fig. 2).

Answer: We have corrected font size of Figure 3b and 3e, and same atom codes are added in Figure 3f.

The reason why we didn’t measure deuterated sample in solid state NMR is that we wanted to see the both dynamics of MDABCO molecule and N-H bond length because ^2H (D) solid state NMR is only sensitive to the dynamics of N-D. If we measure ^2H solid-state NMR with deuterated MNP_3 , we cannot know whether the increased dynamics of deuterium is from rotation of whole MDABCO molecule or from the vibration of N-D. Based on our current data, it is sufficient to conclude that MDABCO molecule is rotating freely in the paraelectric phase because the solid state ^1H NMR result shows sharp lines for all the protons in MDABCO molecules. In addition, we know that N-H bond length is reduced in paraelectric phase by checking the peak shift of the N-H proton.

3. In Fig 4a, why the deuterated sample show significantly reduced polarization compared with the original phase at the same conditions? Any possible reason? The PFM Figs. 4f and 4g do not look perfect, can the authors give some explanation?

Answer: The smaller polarization value of D-MNP₃ (4.6 μC/cm²) than MNP₃ (5.7 μC/cm²) at 5 Hz can be attributed to two factors: first, D-MNP₃ (482.09 Å³) has a larger unit cell volume than MNP₃ (477.13 Å³); secondly, the A site cation displacement of D-MNP₃ (72.56 pm) is shorter than MNP₃ (73.56 pm) due to the Ubbelohde effect. The polarization values at 5 Hz differ by about 23%, but it increases to 84% at 50 Hz where the polarization value of D-MNP₃ is 2.7 μC/cm² and MNP₃ is 5.0 μC/cm². Because the decrease in polarization at a higher frequency is due to the inability of the switching dipoles to keep up with the oscillating field, the decrease in polarization at a higher frequency is more apparent in D-MNP₃ due to a stronger hydrogen bond. We have added the explanation in the revised manuscript on page 13 lines 229-237.

The reason for the non-ideal PFM phase and amplitude switching curves in Fig. 4f and 4g may be due to the polycrystalline sample we used for measurement. Because our sample (MNP₃) has a much higher coercive field (110 kV/cm) than MNI₃ (12 kV/cm) and other molecular ferroelectrics, we cannot obtain PFM phase and amplitude switching curves on single-crystal thicker than 30 μm thick (need more than 300 V exceeding the limit of PFM instrument). Therefore, we made 1 μm thick spray-coated sample to record the switching of the PFM phase and amplitude. Because the spray coated film is polycrystalline, it does not have a uniform thickness (see Supplementary Information in Figure S15), and this makes the switching less ideal.

4. Some format problems: the format of references need to be reorganized to make them consistent: refs 2-5, 10, 12 etc.; the letters in cell parameters and space groups need to be italic etc.

Answer: We corrected the format of references and changed the cell parameters and space groups to be italic.

5. Some important references need to be added: Science 311 (2006) 1270; Nature Materials Reviews, 2017, 2, 16099; APL Materials, 2021, 9, 070401; Wiley book:

<https://onlinelibrary.wiley.com/doi/book/10.1002/9783527344338.<https://ddec1-0-en-ctp.trendmicro.com:443/wis/clicktime/v1/query?url=https%3a%2f%2fonlinelibrary.wiley.com%2fdoi%2fbook%2f10.1002%2f9783527344338.&umid=6112d042-cdfd-422e-90d6-e2f41e4aef88&auth=8d3ccd473d52f326e51c0f75cb32c9541898e5d5-67bb336466ee0cf55a88478e637cad6cfe927492>>

Answer: We have added these references in reference numbers 1, 3, 5, 6 in the revised manuscript.

REVIEWERS' COMMENTS

Reviewer #1 (Remarks to the Author):

Authors have taken care of all the concerns. I am happy with the revision. The article reads well now and it is suitable for publication.

Reviewer #3 (Remarks to the Author):

The authors have addressed all my questions, so I recommend the publication of this paper in its current version

Point by Point response for

Tailoring the Coercive Field in Ferroelectric Metal-Free Perovskites by Hydrogen Bonding

REVIEWER COMMENTS

Reviewer #1 (Remarks to the Author):

Authors have taken care of all the concerns. I am happy with the revision. The article reads well now and it is suitable for publication.

Answer: Thank you so much.

Reviewer #3 (Remarks to the Author):

The authors have addressed all my questions, so I recommend the publication of this paper in its current version.

Answer: Thank you so much.